# Risk Factors for Treatment Toxicity and High Side Effect Burden Among Breast Cancer Survivors: A Retrospective Chart Review

**DOI:** 10.3390/cancers17020328

**Published:** 2025-01-20

**Authors:** Muna Alkhaifi, Elwyn Zhang, Malika Peera, Katarzyna Jerzak, Gregory Czarnota, Andrea Eisen, Amanda Roberts, Carlos Amir Carmona-Gonzalez, Rosanna Pezo, Sonal Gandhi

**Affiliations:** 1Department of Medical Oncology & Hematology, Odette Cancer Centre, Sunnybrook Health Sciences Centre, University of Toronto, Toronto, ON M4N 3M5, Canada; elwyn.zhang@sri.utoronto.ca (E.Z.); katarzyna.jerzak@sunnybrook.ca (K.J.); andrea.eisen@sunnybrook.ca (A.E.); sonal.gandhi@sunnybrook.ca (S.G.); 2Faculty of Health Sciences, Queen’s University, Kingston, ON K7L 3N6, Canada; malika.peera@queensu.ca; 3Department of Radiation Oncology Odette Cancer Centre, Sunnybrook Health Sciences Centre, University of Toronto, Toronto, ON M4N 3M5, Canada; gregory.czarnota@sunnybrook.ca; 4Department of Surgical Oncology Odette Cancer Centre, Sunnybrook Health Sciences Centre, University of Toronto, Toronto, ON M4N 3M5, Canada; amanda.roberts@sunnybrook.ca; 5Division of Medical Oncology, Department of Medicine, Sunnybrook Health Sciences Centre, Toronto, ON M4N 3M5, Canada; carlos.carmona@sunnybrook.ca; 6Department of Biological Sciences, Odette Cancer Research Program, Sunnybrook Research Institute, Toronto, ON M4N 3M5, Canada; rossanna.pezo@sunnybrook.ca

**Keywords:** breast cancer survivors, toxicity, side effects, quality of life, retrospective

## Abstract

Breast cancer survivors face unique challenges and sequelae that can arise due to their disease or treatment and that often persist for many years. Physical, emotional, psychological, and social toxicities are common in breast cancer survivors, which can lead to significant negative impacts on quality of life. While many studies have investigated the relationship between various breast cancer sequelae and quality of life, very few studies have investigated the patient, disease, and treatment factors underlying the likelihood of experiencing side effects in breast cancer survivorship. This retrospective study aims to describe the sequelae, side effects, and toxicities experienced by Canadian breast cancer survivors at a breast cancer survivorship clinic at an academic tertiary cancer centre and to identify potential risk factors which may be associated with increased side effect burden. This will potentially help to predict the extent of a patient’s anticipated need for clinical services, improving the current breast cancer survivorship program’s ability to provide efficient and effective care.

## 1. Introduction

Breast cancer (BC) remains a leading malignancy in terms of incidence among Canadians, with one in eight women expected to be diagnosed in their lifetime [1]. Fortunately, recent advances in early detection and breakthrough innovations in treatment strategies have increased the 5-year survival rate of women with early-stage breast cancer to 90% [2,3]. As a result, the number of breast cancer survivors has increased significantly over the past 15 years, with an estimated overall prevalence of 2% among Canadian women in 2022 [4]. According to the National Cancer Institute, the term “cancer survivor” includes individuals from the time of cancer diagnosis and continues even after the completion of treatment [5]. In general, cancer survivorship care and services are designed to address ongoing issues that persist in the absence of active disease following the completion of initial cancer treatments, such as surgery, chemotherapy, and radiation [6].

Survivors of breast cancer face unique challenges and sequelae that can arise as a result of their disease or its treatment, and these can persist for many years [2]. Physical effects can include chronic pain, lymphedema, fatigue, neuropathy, weight gain, fertility issues, or hot flashes [7,8,9,10]. Emotional, psychological, and social impacts are also common and can include depression, anxiety, fear of recurrence, and body image concerns [11,12,13,14]. For example, a systematic review and meta-analysis found that the prevalence of psychological distress in patients with breast cancers was 50% [15]. Unsurprisingly, the breadth, chronic nature, and severity of potential BC sequelae can lead to a significant negative impact on survivors’ quality of life (QoL) [9,16,17,18]. For example, restrictions in QoL, such as lower scores on emotional, social, physical and functional scales, were found to be significant for BC survivors 10 years after treatment [19,20,21]. Beyond impacting survivors’ QoL, burdensome side effects can also become a barrier to adherence to ongoing treatments. Adjuvant hormone therapy, for example, has been demonstrated to have a tangible negative effect on patient-reported outcomes and QoL, which can in turn lead to decreased adherence to a treatment regimen intended to manage risk of recurrence [22,23]. This in turn can also impact patient outcomes if patients stop prescribed therapies before the ideal timeframe of treatment [24,25]. The impact of these BC sequelae has motivated investigation into risk factors that may predict greater burden. There is a growing body of evidence that younger age, more recent diagnoses, chemotherapy, and endocrine therapy are most closely associated with greater risk of side effects such as chronic pain and sexual dysfunction [26,27,28]. However, while many studies have investigated the relationship between various BC sequelae and QoL, understanding of the patient, disease, and treatment factors underlying the likelihood of experiencing these side effects is less holistic and tends to focus on individual toxicities such as pain, depression, fatigue, and body image issues [11,29,30,31]. The focus is also on outcomes arising from particular interventions; for example, a recent study on the impact of breast reconstruction found that patients who underwent reconstruction had a better QoL, better mental health, and less stress, yet experienced greater physical discomfort [32]. Nonetheless, research remains limited on the unique side effects experienced by younger and older breast cancer survivors [33,34].

There is a need for comprehensive survivorship care that addresses the long-term needs of breast cancer survivors across multiple realms [35,36]. The cancer survivorship framework developed by Nekhlyudov et al. highlights key areas of quality cancer survivorship care: surveillance and management of physical, psychosocial, and chronic medical conditions; prevention and surveillance for recurrence and new cancers; and health promotion and disease prevention [37]. Furthermore, according to the American Society of Clinical Oncology (ASCO) guidelines, BC survivorship care requires a comprehensive, holistic approach that considers monitoring for recurrence as well as screening and managing long-term side effects [23].

Many models have been developed to address the complex needs of cancer survivors [38]. Oncologist-led survivorship care is the most prevalent model of survivorship care; however, oncologists often struggle to maintain long-term relationships with cancer survivors [39,40,41]. As such, this model shows limited sustainability and fails to address survivors’ unique physical, psychosocial, supportive, informational, and rehabilitative needs [42]. Furthermore, the growing number of cancer survivors has overwhelmed the traditional oncology-led survivorship model, resulting in the shift of survivorship care to the primary care domain [43,44]. Primary care providers play a critical role in enhancing the well-being and satisfaction of cancer survivors, and patients have reported equivalent or higher satisfaction with primary care-led models compared to other survivorship models [38,45].

The patient population of the Sunnybrook Breast Cancer Survivorship Program is referred by medical oncologists at the Louise Termerty Breast Cancer Centre. This includes women who have completed surgery, radiation, and intravenous chemotherapy or targeted treatments. Only women who have no current evidence of disease are referred to this clinic. Many patients are still on anti-hormone treatments and/or bisphosphonate therapy, often requiring maintenance adjuvant therapies for several years (5 to 10), which can cause ongoing side effects and impact QoL.

This study describes the findings from a retrospective chart review of patients referred to and seen in the Sunnybrook Breast Survivorship Clinic. The purpose of this review is to describe the sequelae, side effects, and toxicities experienced by this cohort, as well as to identify potential risk factors leading to increased side effect burden, likelihood of experiencing physical or psychological sequelae, and greater health service utilization. This will potentially help to predict the extent of a patient’s anticipated need for clinical services and would improve the program’s ability to provide efficient and effective care. Furthermore, this review aims to describe the experience of survivors at our clinic to add to the existing literature on specific high-risk populations and to determine any additional trends in our specific population.

## 2. Materials and Methods

A single-centre retrospective chart review was undertaken of patients who were treated at the Sunnybrook Breast Cancer Survivorship Clinic from its establishment on 6 July 2022, through 30 September 2023 (*n* = 435). Patients were included if they were older than 18 years, had completed treatment for at least one breast cancer at the time of extraction, and had attended at least one appointment at the Sunnybrook Breast Cancer Survivorship Clinic. Patients were excluded if they did not attend any appointments; had not had a confirmed diagnosis of breast cancer; or were undergoing surgery, chemotherapy, or radiation therapy for their first breast cancer at the time of data extraction. The Sunnybrook Health Sciences Centre Research Ethics Board reviewed and approved this study (REB number: 6008). Consent was waived by the REB as patient data was unidentified during chart review. Eligible patients’ hospital electronic medical records were reviewed, and key information was extracted, such as demographics, symptoms, treatment history, and any reported side effects of cancer treatment.

### 2.1. Data Collection

As part of the survivorship clinic care, patients are comprehensively and proactively screened for symptom burden (e.g., depression, anxiety, joint pain) during their first visit in accordance with ASCO guidelines for breast cancer survivorship care [46]. All data are documented in patients’ electronic medical records (EMRs), which were subsequently utilized for chart reviews to extract relevant parameters. Any future records of follow-up appointments with the program were also included for symptom data collection.

### 2.2. Analysis

#### Outcome Variables

Data were deidentified and analysis was performed using R version 4.3 (R Foundation for Statistical Computing: Vienna, Austria) [47]. Categorical variables were summarized using counts/percentages, and continuous variables were summarized using means/SDs. Binary secondary variables were derived from the primary data. High side effect burden was operationalized as having at least two side effects impacting quality of life. This was determined from previous literature [48] and team consensus. Patients with two or more side effects as identified during the extraction step were assigned a 1. All other patients were assigned a 0. Side effects were grouped broadly into psychological (e.g., depression, anxiety, body image issues) and physical (e.g., joint pain, dyspareunia, hot flashes) in accordance with ASCO guidelines for breast cancer survivorship care [46]. Patients with at least one side effect falling under the psychological or physical category were assigned a 1 for the respective variable or 0 otherwise. Patients were considered to have required close follow-up if they had more than one appointment scheduled per year for a reason other than routine imaging, such as side effect management, patient request, or changes in medication. ASCO guidelines generally recommend routine physical examinations be undertaken every 3 to 6 months for the first 3 years after primary therapy, every 6 to 12 months between 3 and 5 years, and annually thereafter [46]. Thus, patients who required more frequent appointments for non-routine reasons were assigned a 1 for this variable and operationalized as requiring close follow-up. All other patients (i.e., those who were seen only annually or only for routine imaging) were assigned a 0. Study team members selected variables for inclusion in bivariate models prior to analysis based on available data, existing literature, and clinical experience.

### 2.3. Statistical Analysis

The magnitude/direction of association between demographic/clinical variables was estimated using both bivariate and multivariable logistic regression models. Estimated odds ratios, 95% confidence intervals, and *p*-values from these models were reported. Determination of variables for inclusion in final multivariable logistic regression models was conducted using bivariate screening and a *p* < 0.20 cut-off. Models were constructed and analysis was performed using R version 4.3 [47]. A *p*-value threshold of 0.05 was used for determination of statistical significance.

## 3. Results

### 3.1. Descriptive Results

Descriptive data of the patient cohort can be found in Table 1. Of 435 patients, all (100%) were female. The mean age of the cohort was 63 (SD 11.65), skewing older, with very few patients under 45. The majority (88%) of patients were postmenopausal, either due to age or induced by hormonal therapy. A patient’s postmenopausal status was established through their medical history and confirmed during the initial appointment. Most patients were between 3 and 10 years from their initial diagnosis of breast cancer, with the mean being 61 months (SD 33.39) (Figure 1). All patients (100%) had surgical treatment of their breast cancer, with 67.4% having received a lumpectomy, 36.2% having received a mastectomy, and 3.6% having received both at some point. Additionally, 15.4% of patients received breast reconstruction, either immediate or delayed (Table 2). The majority of patients received adjuvant radiotherapy (86.7%) and adjuvant endocrine therapy (83.4%). Just over half (52%) of patients received chemotherapy. A total of 61% of clinic patients were on active endocrine therapy at the time of chart review (Table 3). Furthermore, 21.8% of patients discontinued a drug due to side effects, most commonly adjuvant endocrine therapies.

### 3.2. Side Effect Burden

The side effect burden is presented in Table 4 and Table 5. The majority of patients (72.6%) reported at least one side effect impacting their quality of life (Table 4), with a smaller majority (55.4%) reporting two or more such effects (Table 5). The most common symptoms experienced by patients were anxiety (29.4%), chronic pain (23.9%), hot flashes (21.4%), and fear of recurrence (19.8%). Notably, 33.8% of patients reported psychological issues such as depression, body image concerns, and severe anxiety. Older age was strongly correlated with a lower likelihood of experiencing greater side effect burden as operationalized by having two or more side effects beginning at the age of 55 and older (Table 6, Figure 2). This correlation was both strong and consistent across bivariate and multivariate models. Additionally, while no significant difference was found between patients who had never smoked and those who had smoked in the past, current smokers were found to be likelier than never-smokers to have a high side effect burden.

Bivariate analysis additionally showed that post-menopausal women were less likely to have a high side effect burden than pre-menopausal women. Notably, patients who had taken aromatase inhibitors (Ais) were less likely than those who had never taken any kind of maintenance hormone therapy to have a high burden. Additionally, survivors with triple negative (TN) disease, as well as those who had undergone chemotherapy, were likelier to experience high side effect burden according to their respective bivariate models. However, neither receptor status nor history of chemotherapy were found to be significant once other variables were introduced through the multivariable model. Interestingly, the results demonstrated that duration from the time of diagnosis did not significantly affect side effect burden.

### 3.3. Physical Side Effects

Physical BC sequelae such as joint stiffness, chronic pain, and dyspareunia were found to be less prevalent among older patients in bivariate models, with the 65–75 and 75 and older age groups being less likely to report such side effects. However, unlike for other outcome measures, no significant correlation between physical sequelae and age persisted in the multivariate analysis (Table 7). Similarly, while post-menopausal women were less likely to have physical side effects in a bivariate model, no significance was found for menopausal status when other variables were introduced. In terms of treatment history, the bivariate analysis found that patients who took AIs were less likely than those who had no hormone maintenance therapy to have physical sequelae. In contrast, patients who had chemotherapy were found to have greater likelihood of physical side effects than those who did not have chemotherapy. Neither of these correlations remained significant in the multivariate model. Notably, patients whose latest disease was a recurrence of a previous diagnosis of BC were significantly less likely to report physical sequelae, a finding replicated in both bivariate and multivariate analyses (OR 0.23 and OR 0.13, respectively).

### 3.4. Psychological Side Effects

A number of clinical and demographic factors were found to influence the likelihood of experiencing psychological side effects. Compared to patients under 45, bivariate modeling found all older age categories to be less likely to experience psychological sequelae such as depression, anxiety, or issues with body image. However, the multivariate model indicated significance only for the 65–75 and 75 and older age groups (Table 8). Similar to the previous outcomes, post-menopausal survivors were found to be less likely to have psychological BC sequelae in a bivariate model, but this did not pass the threshold of significance in a multivariate model. Additionally, current smokers were significantly likelier to experience psychological symptoms than patients who had never smoked in both bivariate and multivariate models (ORs of 4.86 and 7.26, respectively).

Disease receptor status was relevant: bivariate modeling found both TP and TN disease to be associated with a greater likelihood of psychological sequelae compared to HR+/HER2- patients. In the multivariate model, those with TP status were still found to be at greater risk (OR 2.72), while TN status was found to not be significant. Bivariate modeling found surgical treatment to be a significant factor for psychological side effects. Notably, patients who had undergone mastectomy, sentinel lymph node (SN) biopsy, or reconstructive surgery were likelier than those who had not undergone each of these interventions to develop psychological sequelae. However, these effects did not persist in a multivariate model. Hormonal therapy and chemotherapy were also of interest in a bivariate model. Patients who had taken AIs were less likely than those without any maintenance hormonal therapy to have psychological side effects (OR 0.39). In contrast, those who underwent chemotherapy were at greater risk (OR 1.67). As with surgical history, neither of these correlations remained significant in a multivariate model.

### 3.5. Clinic Interventions

All patients (100%) received survivorship care in accordance with the latest guidelines, including the ASCO Breast Cancer Survivorship Guideline (2016) [46], the National Comprehensive Cancer Network (NCCN) Breast Cancer Guideline (2024) [49] and the National Cancer Institute (NCI) “Quality of Cancer Survivorship Care Framework” by Nekhlyudov et al. (2019) [37]. However, a further 39% required specific interventions or counseling due to moderate or severe symptoms and side effects. While the majority of patients were seen annually, over a third (39%) required close follow-up due to their symptoms. Additionally, 13.6% of patients required psychological support as part of their clinic care. The majority of patients seen since the clinic’s establishment were not yet discharged at the time of extraction (85%). Four percent of patients were referred back to their medical oncologist, most commonly for recurrence of disease. A further 11% were offered discharge; of these, 46% declined. Ultimately, 6% of patients were discharged to primary care.

Additionally, age group was found to be the most relevant factor for whether a patient required close follow-up (and thus greater clinic service utilization). Compared to patients under 45, bivariate analysis found patients in the 45–55, 55–65, 65–75, and 75 and older age groups to be significantly less likely to require close follow-up, a correlation that remained strong in the multivariate model. The bivariate analysis also found that post-menopausal patients were less likely to require close follow-up than pre-menopausal patients, though as with other outcome measures, this correlation did not remain significant in a multivariate model (Table 9).

## 4. Discussion

This retrospective chart review aimed to estimate the prevalence of BC sequelae and their associated burden (operationalized as having two or more identified side effects) in a cohort of adult breast cancer survivors from a single academic cancer center in Toronto, Canada, as well as to identify potential disease and patient factors associated with higher symptom burden. Although the literature presents a high side effect burden as one or more side effects [48], since 72.6% of patients presented with at least one side effect, the team decided to characterize a high side effect burden as two or more side effects to better understand the potential risks to breast cancer survivors. The majority (55.4%) of patients at the studied centre were found to have two or more side effects, with a particularly high incidence of anxiety, chronic pain, hot flashes, and general fear of recurrence. Older age was correlated with a lesser likelihood of a high side effect burden, lesser likelihood of experiencing psychological sequelae, and a lower likelihood of requiring close follow-up.

### 4.1. Treatment Related Factors

Chemotherapy was found in the bivariate analysis to be associated with a greater likelihood of a high side effect burden (*p* < 0.01). Chemotherapy is known to cause various long-term side effects in breast cancer patients, including fatigue, peripheral neuropathy, cognitive impairment (such as brain fog), and fertility issues [50]. This finding is in line with the available literature on the impact of chemotherapy on quality of life, symptoms, and long-term experience, emphasizing the importance of quality follow-up care and proactive screening for these patients [51,52,53].

The bivariate analysis also found that patients taking aromatase inhibitors (AIs) were less likely to report a high symptom burden when compared to those taking no anti-estrogen therapies. This result is unexpected given the extensive evidence that AIs have a generally negative effect on QoL for a number of measures [54,55]. Furthermore, a previous retrospective chart review of breast cancer patients prescribed with AIs found that 82% of 179 women prescribed AIs had at least one symptom or side effect described in their medical records [56]. As this finding was not corroborated by the multivariate model, it is hard to draw definitive conclusions from this statistic, which is possibly confounded through association with factors such as prior treatments and time from diagnosis.

### 4.2. Patient-Related Factors

Bivariate analyses also demonstrated a number of interesting associations that remained robust in a multivariate model. Age was strongly associated with all outcomes of interest, with younger age being predictive of greater likelihood of a high side effect burden (including psychological symptoms) and greater likelihood of requiring close follow-up. This result is particularly notable as previous research has indicated older patients to be higher need, with greater comorbidities and high impact on quality of life [57,58,59]. In a recent Canadian study, over 70% of cancer survivors over 75 years old reported comorbidities, and 68.2% reported physical challenges including side effects, physical capacity, and changes in body function or appearance [60]. This discrepancy may arise from population-level differences in our cohort or from differences in reporting: while the previous survey asked for patients to report all of their symptoms, in this retrospective chart review, patients may not have mentioned experiencing similar challenges if they did not deem them as relevant to their survivorship care or treatment toxicities. However, this is not to say that young survivors do not have unique and pressing challenges, with significant impact on QoL through both physical and psychological barriers. For example, the potential early onset of menopause, physiological changes to the body, impact on fertility, body image concerns, and emotional trauma can cause particular distress for younger patients [61]. Younger patients have also been shown to experience significantly greater fear of recurrence than older survivors, which in turn may lead to greater usage of healthcare services [62,63].

The findings generally support the perspective that young survivors have unique needs. Thankfully, dedicated support does exist for younger breast cancer survivors in the form of young survivorship programs which are tailored to this population [64]. Notably, however, these programs often use conventional age thresholds of 40 or 45. The analysis found no significant difference in measured outcomes between those under 45 and those under 55—perhaps indicating a need for dedicated support among young survivors that are somewhat older than the conventional threshold.

While active smoking is known to be moderately associated with an increase in the risk of breast cancer, as well as second primary cancers among survivors, its relationship to BC sequelae and QoL in survivorship is somewhat less clear [65,66]. Being a current smoker was strongly associated with a number of negative outcomes at significant levels, both in the bivariate and multivariate models. For example, compared to patients who had never smoked or those who had smoked in the past, smokers were much likelier to report a high side effect burden (OR 4.01, *p* = 0.005). The reason for this relationship is not clear and is difficult to interrogate through a retrospective chart review. A 2013 study found that BC survivors who were former smokers had an exaggerated and prolonged inflammatory response to a laboratory stressor relative to patients who had not smoked. The authors hypothesized this could indicate hypothalamic-pituitary-adrenal dysregulation among smokers and thus a modified or worsened response to stressors normally associated with BC survivorship [67]. It is possible that there are confounding variables not captured by this study’s methods that drive this effect. For example, prior studies about smoking among cancer survivors found a complex array of reasons to continue smoking, ranging from poor socioeconomic and psychological support to high nicotine dependence [68,69]. Many of these factors were not accounted for during this retrospective chart review and may have influenced the results.

Nearly half (46%) of patients offered discharge and transition to family physicians declined, preferring to remain in follow-up. This is a considerable barrier to the objective of facilitating transition to community care for patients otherwise considered to be low-risk. While it is generally understood that patients often prefer specialist follow-up due to concerns about primary care physicians’ familiarity with BC and monitoring for recurrence, transition to primary care is considered to be a desirable ultimate goal of specialized cancer survivorship care [70,71,72]. Previous studies have found that BC survivors may form relatively strong relationships with their specialist care providers and derive significant reassurance from specialty care and associated surveillance [73]. Further, survivors did not conceptualize their primary care providers as having a significant role in survivorship care, viewing BC survivorship as distinct from other healthcare concerns. This could lead to further reluctance to be discharged from specialist care and transition to a PCP-led model [70,73].

### 4.3. Limitations

Due to the limitations of retrospective chart reviews, patients were reported as experiencing a side effect if their medical records indicated so at any point following their treatment for BC. It is possible that some patients experienced these side effects for reasons unrelated to their treatment or experienced a lesser or greater quality of life impact than other patients. Outcome measures are another potential limitation of this study. In particular, the operationalization of a high side effect burden as having two or more side effects is inexact and does not directly measure the true impact of these sequelae on patients’ lives. Further, the Sunnybrook Breast Cancer Survivorship Program patient cohort is not necessarily representative of the broader population of breast cancer survivors. Patients that are referred to the clinic may differ systematically in side effect burden, need for intervention, or other variables. These limitations complicate analysis and generalizability. However, despite these limitations, the present study offers key insights into the deep need among breast cancer survivors for ongoing management of side effects. That the relationships between young age, smoking, and side effect burden remained robust through statistical testing also reveals new avenues for investigation.

### 4.4. Future Directions

The findings of this study highlight a significant burden of side effects experienced by patients, including psychological effects and chronic pain. To address this, additional resources should be allocated at the clinic, such as a nurse navigator and psychological support services, including counselling and peer support programs. Additionally, multidisciplinary and personalized teams should be employed to support patients in managing the side effects of breast cancer treatment. Moreover, the integration of virtual care options can facilitate more frequent follow-up, thereby alleviating the substantial burden of side effects experienced by survivors. Additionally, providing patients with comprehensive educational resources may assist them in navigating and managing the various side effects of breast cancer.

Given that women ages 45–55 experience a comparable burden of side effects compared to those under 45, these resources should be tailored specifically to support women under the age of 55. Moreover, targeted support programs can be developed for this demographic, in alignment with the study’s findings, including specialized personal support programs and dedicated multidisciplinary teams.

Furthermore, the results indicated that survivors who smoked were at a higher risk of experiencing a side effect burden, underscoring the unique needs of this population. These findings highlight the importance of actively referring smoking survivors to the smoking cessation program at the Odette Cancer Centre to support their cessation efforts and enhance their recovery.

Additionally, a high patient reluctance to be discharged is notable. Patients offered discharge are those who clinicians believe have well-managed symptoms, an effective plan for monitoring for recurrence, and would do well in primary care. That so many of these patients declined and expressed a desire to remain in follow-up care is a significant barrier to one of the main pillars of survivorship care. While common reasons for this reluctance are broadly understood, such as a desire to remain in specialist care and concerns about the oncology knowledge of primary care physicians, there remains a paucity of evidence for the most effective ways to reassure patients of the effectiveness and importance of primary care transitions. Future research, specifically qualitative studies, should investigate the nature of patients’ reluctance and how to best address their concerns.

These findings reveal a number of potential future research directions. In particular, while the association between younger age and greater burden is generally concordant with the literature, it is notable that no statistically significant difference was found between the under 45 and 45–55 age categories. This could indicate that patients over the conventional “young survivorship” cutoff age may experience similar challenges to those generally considered young survivors. Future research could investigate whether survivors under 55 experience similar health concerns and if their inclusion in programs geared towards young survivorship issues is warranted. Additionally, prospective research initiatives are needed to validate the findings of this study, particularly concerning smoking and younger breast cancer survivors, to further understand the unique needs of these demographic groups.

## 5. Conclusions

There are significant clinical implications from the findings of this study. In particular, the study’s findings indicate an acute need among younger breast cancer survivors, who are more likely than older patients to report symptoms impacting their quality of life. In particular, psychological side effects such as anxiety, fear of recurrence, and body image issues are prominent concerns among these patients. Further, while it is generally understood that smoking is associated with poorer health outcomes both among survivors and the general population, this study’s specific findings open the door for future investigation and add to the clinical evidence for the importance of smoking cessation for survivors. The relationship cannot be said to be causal, however, and smoking cessation is complicated by a number of socioeconomic and personal factors. In general, practitioners should focus on individual patient factors that could influence success in smoking cessation.

Overall, the present study offers several insights into the needs of a dedicated breast cancer survivorship clinic’s patient cohort. Future research should seek to further investigate these relationships and determine whether they indicate a need to revise currently accepted guidelines for survivorship care or to direct specific consideration to particular subgroups of survivors. Further, understanding the nature and source of patients’ reluctance to be discharged is a critical next step to delivering effective transitional care for breast cancer survivors into the future.

## Figures and Tables

**Figure 1 cancers-17-00328-f001:**
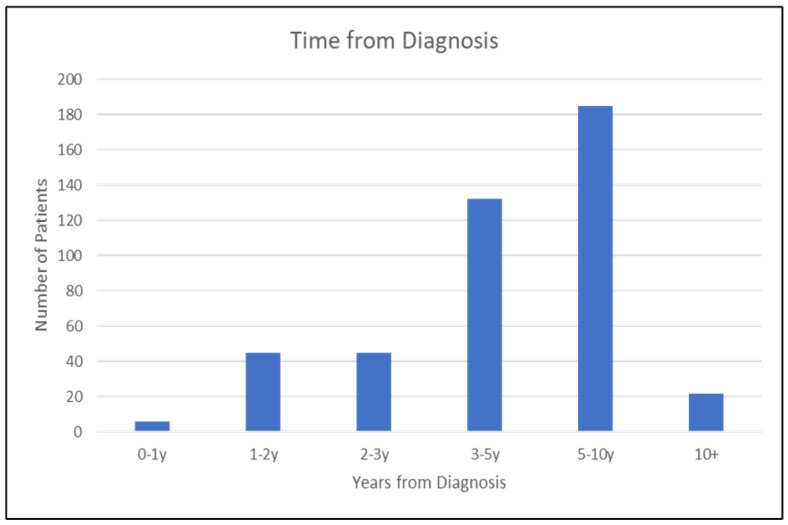
Clinic population time from last BC diagnosis.

**Figure 2 cancers-17-00328-f002:**
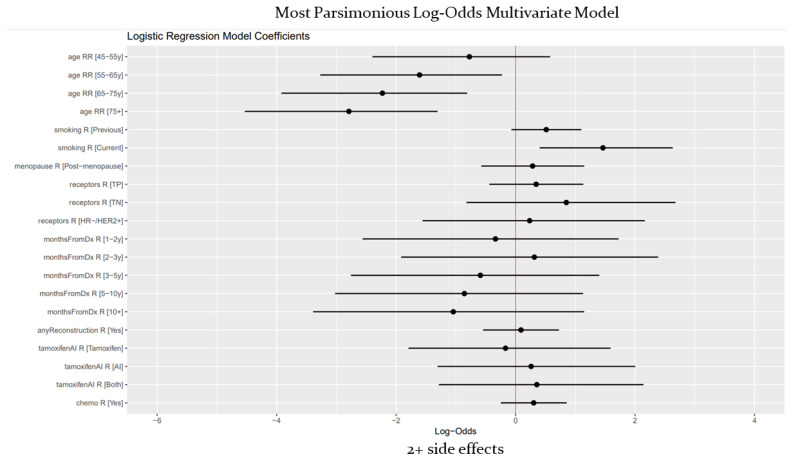
Multivariate log-odds model (two or more side effects).

**Table 1 cancers-17-00328-t001:** Descriptive statistics.

Characteristic	Overall (*n* = 435)
**Age Category (%)**	
0–45 years	15 (3.4%)
45–55 years	101 (23.2%)
55–65 years	122 (28.0%)
65–75 years	110 (25.3%)
75+	87 (20.0%)
**Smoking Status**	
Never	336 (77.6%)
Previous	76 (17.6%)
Current	21 (4.8%)
**Alcohol Usage**	
None	281 (65.3%)
2 or fewer drinks per week	85 (19.8%)
3 or more drinks per week	64 (14.9%)
**Menopausal Status**	
Pre-menopause	51 (12.0%)
Post-menopause	375 (88.0%)
**Nodal Status**	
No	273 (62.8%)
Yes	162 (37.2%)
**Receptor Status**	
HR+/HER2-	316 (73.8%)
TP	44 (10.3%)
TN	42 (9.8%)
HR-/HER2+	26 (6.1%)
**Laterality**	
Left	215 (49.4%)
Right	201 (46.2%)
Bilateral	19 (4.4%)
**Previous Breast Cancer**	
No	405 (93.1%)
Yes	30 (6.9%)
**Current Cancer Recurrent**	
No	425 (97.7%)
Yes	10 (2.3%)
**Years from Dx (%)**	
0–1 years	6 (1.4%)
1–2 years	45 (10.3%)
2–3 years	45 (10.3%)
3–5 years	132 (30.3%)
5–10 years	185 (42.5%)
10+	22 (5.1%)
**Currently on Endocrine Therapy (%)**	
No	170 (39.1%)
Yes	265 (60.9%)

**Table 2 cancers-17-00328-t002:** Surgery types.

Surgery	Total (*n* = 435)
Lumpectomy	292 (67.4%)
Mastectomy	157 (36.2%)
SN Biopsy	355 (81.6%)
Axillary Dissection	58 (13.3%)
Reconstruction	67 (15.4%)

**Table 3 cancers-17-00328-t003:** Treatment history.

Treatment History	Total (*n* = 435)
Chemotherapy	225 (52.0%)
Radiation	377 (86.7%)
Endocrine Therapy	363 (83.4%)
Tamoxifen	84 (19.3%)
AI	188 (43.2%)
Both	91 (20.9%)
Zoledronic Acid	163 (37.5%)

**Table 4 cancers-17-00328-t004:** Side effects experienced by patients.

Side Effect	Number of Patients Reporting
	(*n* = 435)
**At Least One Side Effect**	**316 (72.6%)**
Anxiety	128 (29.4%)
Chronic Pain	104 (23.9%)
Hot Flashes	93 (21.4%)
Fear of Recurrence	86 (19.8%)
Joint Pain	83 (19.1%)
Dry Vagina	81 (18.6%)
Lymphedema	59 (13.6%)
Depression	54 (12.4%)
Fatigue	41 (9.4%)
Issues Relating to Body Image	37 (8.5%)
Low Sex Drive	35 (8.0%)
Dyspareunia	34 (7.8%)
Neuropathy	32 (7.4%)
Work/Social/Family Issues	29 (6.7%)
Skin Changes	29 (6.7%)
Weight Gain	27 (6.2%)
Brain Fog	25 (5.7%)
Barrier to Physical Functioning	21 (4.8%)
Fertility Issues	5 (1.1%)
No Side Effects	119 (27.4%)

**Table 5 cancers-17-00328-t005:** Descriptive statistics (outcomes).

Outcome	Overall
**Has 2 or More Side Effects**	
No	194 (44.6%)
Yes	241 (55.4%)
**Has Physical Sequelae**	
No	135 (31.0%)
Yes	300 (69.0%)
**Has Psychological Sequelae**	
No	298 (68.5%)
Yes	137 (31.5%)
**Required Close Follow-up**	
No	264 (60.7%)
Yes	169 (39.3%)

**Table 6 cancers-17-00328-t006:** Odds ratios, confidence intervals, and *p*-values from bivariate and multivariate logistic regression models quantifying the association between demographic/clinical features and the likelihood of experiencing two or more side effects.

Variable		Bivariate Model				Multivariate Model			
		Odds Ratio	95% Confidence Interval		*p*-Value	Odds Ratio	95% Confidence Interval		*p*-Value
			Lower	Upper			Lower	Upper	
Age	0–45 years	NA	NA	NA	NA	NA	NA	NA	NA
	45–55 years	0.19	0.04	0.64	**0.0136**	0.16	0.03	0.57	**0.009**
	55–65 years	0.19	0.04	0.64	**0.0139**	0.18	0.04	0.68	**0.0181**
	65–75 years	0.11	0.02	0.36	**0.001**	0.1	0.02	0.37	**0.0014**
	75+	0.12	0.03	0.42	**0.0020**	0.13	0.03	0.48	**0.0047**
Smoking	Never	NA	NA	NA	NA	NA	NA	NA	NA
	Previous	1.43	0.86	2.36	0.1686	1.49	0.84	2.62	0.1662
	Current	2.35	0.97	5.9	0.061	2.18	0.84	5.79	0.1106
Alcohol Use	None	NA	NA	NA	NA	-	-	-	-
	≤2/weeks	1.3	0.79	2.12	0.2954	-	-	-	-
	>2/weeks	0.73	0.4	1.29	0.2898	-	-	-	-
Menopausal Status	Pre-menopause	NA	NA	NA	NA	NA	NA	NA	NA
	Post-menopause	0.47	0.26	0.85	**0.0125**	0.8	0.36	1.76	0.5737
T-Stage	T1	NA	NA	NA	NA	-	-	-	-
	T2	1.34	0.86	2.07	0.1901	-	-	-	-
	T3	1.8	0.85	3.81	0.1194	-	-	-	-
	T4	1.08	0.22	4.51	0.9161	-	-	-	-
	Ti	1.16	0.47	2.75	0.7424	-	-	-	-
Nodal Status	No	NA	NA	NA	NA	-	-	-	-
	Yes	0.97	0.65	1.44	0.8643	-	-	-	-
Receptor Status	HR+/HER2-	NA	NA	NA	NA	-	-	-	-
	TP	1	0.52	1.9	0.9897	-	-	-	-
	TN	0.98	0.5	1.89	0.9561	-	-	-	-
	HR-/HER2+	1.6	0.71	3.59	0.2537	-	-	-	-
Previous Breast Cancer	No	NA	NA	NA	NA	-	-	-	-
	Yes	0.77	0.34	1.65	0.5084	-	-	-	-
Current Cancer Recurrent	No	NA	NA	NA	NA	-	-	-	-
	Yes	0.66	0.14	2.42	0.5563	-	-	-	-
Time from Diagnosis	0–1 years	NA	NA	NA	NA	NA	NA	NA	NA
	1–2 years	1.05	0.18	6.18	0.9592	0.58	0.07	4	0.5767
	2–3 years	1.81	0.3	10.82	0.4962	1.16	0.14	8.14	0.8823
	3–5 years	0.59	0.11	3.3	0.5285	0.36	0.04	2.32	0.2791
	5–10 years	0.46	0.08	2.57	0.3559	0.23	0.03	1.46	0.1179
	10+	0.47	0.07	3.08	0.4155	0.26	0.03	2.02	0.2012
Lumpectomy	No	NA	NA	NA	NA	-	-	-	-
	Yes	0.73	0.49	1.1	0.135	-	-	-	-
Mastectomy	No	NA	NA	NA	NA	-	-	-	-
	Yes	1.42	0.95	2.12	0.0874	1.3	0.77	2.19	0.3298
SN Biopsy	No	NA	NA	NA	NA	-	-	-	-
	Yes	1.01	0.62	1.68	0.9547	-	-	-	-
Axillary Dissection	No	NA	NA	NA	NA	-	-	-	-
	Yes	1.03	0.58	1.8	0.9165	-	-	-	-
Reconstruction	No	NA	NA	NA	NA	NA	NA	NA	NA
	Yes	1.65	0.98	2.8	0.0599	1.15	0.56	2.35	0.7083
Endocrine Therapy Type	Neither	NA	NA	NA	NA	-	-	-	-
	Tamoxifen	1.25	0.66	2.39	0.4969	-	-	-	-
	AI	0.85	0.48	1.5	0.5642	-	-	-	-
	Both	1.52	0.81	2.89	1.1904	-	-	-	-
Chemotherapy	No	NA	NA	NA	NA	-	-	-	-
	Yes	1.38	0.93	2.03	0.1072	-	-	-	-
Zoledronic Acid	No	NA	NA	NA	NA	-	-	-	-
	Yes	0.8	0.54	1.2	0.2874	-	-	-	-
Radiation	No	NA	NA	NA	NA	-	-	-	-
	Yes	0.79	0.45	1.4	0.4232	-	-	-	-
Currently on Endocrine Therapy	No	NA	NA	NA	NA	-	-	-	-
	Yes	1.08	0.73	1.61	0.6922	-	-	-	-

NA refers to data “not available”.

**Table 7 cancers-17-00328-t007:** Odds ratios, confidence intervals, and *p*-values from bivariate and multivariate logistic regression models quantifying the association between demographic/clinical features and the likelihood of experiencing physical side effects.

Variable		Bivariate Model				Multivariate Model			
		Odds Ratio	95% Confidence Interval		*p*-Value	Odds Ratio	95% Confidence Interval		*p*-Value
			Lower	Upper			Lower	Upper	
Age	0–45 years	NA	NA	NA	NA	NA	NA	NA	NA
	45–55 years	0.44	0.07	1.75	0.3056	0.44	0.05	2.09	0.3524
	55–65 years	0.35	0.05	1.36	0.185	0.36	0.05	1.76	0.2568
	65–75 years	0.19	0.03	0.74	**0.0349**	0.19	0.02	0.95	0.0684
	75+	0.16	0.02	0.61	**0.0192**	0.22	0.03	1.15	0.104
Smoking	Never	NA	NA	NA	NA	NA	NA	NA	NA
	Previous	1.02	0.61	1.72	0.9532	1.08	0.61	1.96	0.7861
	Current	3.55	1.17	15.39	**0.0454**	3.11	0.95	14.12	0.0886
Alcohol Use	None	NA	NA	NA	NA	-	-	-	-
	≤2/weeks	1.52	0.91	2.6	0.1172	-	-	-	-
	>2/weeks	1.3	0.74	2.34	0.3739	-	-	-	-
Menopausal Status	Pre-menopause	NA	NA	NA	NA	NA	NA	NA	NA
	Post-menopause	0.5	0.25	0.97	**0.0488**	0.9	0.35	2.26	0.8169
T-Stage	T1	NA	NA	NA	NA	-	-	-	-
	T2	1.38	0.88	2.17	0.1632	-	-	-	-
	T3	1.99	0.89	4.9	0.1077	-	-	-	-
	T4	1.99	0.45	13.8	0.4042	-	-	-	-
	Ti	1.25	0.52	3.2	0.6307	-	-	-	-
Nodal Status	No	NA	NA	NA	NA	-	-	-	-
	Yes	1.06	0.71	1.6	0.7616	-	-	-	-
Receptor Status	HR+/HER2-	NA	NA	NA	NA	NA	NA	NA	NA
	TP	1.23	0.64	2.45	0.5372	1	0.44	2.33	0.993
	TN	2.04	1	4.51	0.0609	1.49	0.24	9.62	0.6644
	HR-/HER2+	2.68	1.06	8.18	0.0539	2.22	0.31	18.04	0.4356
Previous Breast Cancer	No	NA	NA	NA	NA	-	-	-	-
	Yes	0.72	0.34	1.56	0.3938	-	-	-	-
Current Cancer Recurrent	No	NA	NA	NA	NA	NA	NA	NA	NA
	Yes	0.23	0.05	0.85	**0.0372**	0.13	0.02	0.62	**0.0187**
Time from Diagnosis	0–1 years	NA	NA	NA	NA	NA	NA	NA	NA
	1–2 years	0.55	0.03	3.89	0.6019	0.86	0.04	7.44	0.9035
	2–3 years	0.8	0.04	5.84	0.8471	1.24	0.06	11.24	0.8629
	3–5 years	0.33	0.02	2.11	0.3153	0.49	0.02	3.82	0.548
	5–10 years	0.27	0.01	1.71	0.2341	0.3	0.01	2.33	0.3092
	10+	0.53	0.03	4.31	0.599	0.67	0.03	6.98	0.7536
Lumpectomy	No	NA	NA	NA	NA	-	-	-	-
	Yes	0.84	0.55	1.28	0.4173	-	-	-	-
Mastectomy	No	NA	NA	NA	NA	-	-	-	-
	Yes	1.13	0.75	1.71	0.5613	-	-	-	-
SN Biopsy	No	NA	NA	NA	NA	-	-	-	-
	Yes	0.82	0.49	1.37	0.4594	-	-	-	-
Axillary Dissection	No	NA	NA	NA	NA	NA	NA	NA	NA
	Yes	1.73	0.95	3.33	0.0842	2.02	0.98	4.35	0.0622
Reconstruction	No	NA	NA	NA	NA	-	-	-	-
	Yes	1.29	0.75	2.29	0.3716	-	-	-	-
Endocrine Therapy Type	Neither	NA	NA	NA	NA	NA	NA	NA	NA
	Tamoxifen	0.77	0.38	1.52	0.454	0.94	0.16	5.6	0.9434
	AI	0.48	0.26	0.85	**0.0138**	0.85	0.15	4.76	0.8454
	Both	1.02	0.51	2.02	0.9655	1.43	0.23	8.73	0.6929
Chemotherapy	No	NA	NA	NA	NA	NA	NA	NA	NA
	Yes	1.73	1.16	2.58	**0.0068**	1.18	0.66	2.11	0.5825
Zoledronic Acid	No	NA	NA	NA	NA	-	-	-	-
	Yes	0.99	0.66	1.49	0.972	-	-	-	-
Radiation	No	NA	NA	NA	NA	-	-	-	-
	Yes	1.41	0.8	2.46	0.2339	-	-	-	-
Currently on Endocrine Therapy	No	NA	NA	NA	NA	-	-	-	-
	Yes	1.12	0.75	1.66	0.5886	-	-	-	-

NA refers to data “not available”.

**Table 8 cancers-17-00328-t008:** Odds ratios, confidence intervals, and *p*-values from bivariate and multivariate logistic regression models quantifying the association between demographic/clinical features and the likelihood of experiencing psychological side effects.

Variable	Bivariate Model					Multivariate Model			
	Odds Ratio		95% Confidence Interval		*p*-Value	Odds Ratio	95% Confidence Interval		*p*-Value
			Lower	Upper			Lower	Upper	
Age	0–45 years	NA	NA	NA	NA	NA	NA	NA	NA
	45–55 years	0.23	0.05	0.76	**0.0279**	0.22	0.04	0.91	0.0514
	55–65 years	0.16	0.03	0.52	**0.0058**	0.15	0.03	0.65	0.0514
	65–75 years	0.07	0.01	0.24	**0.0001**	0.06	0.01	0.27	**0.0006**
	75+	0.02	0	0.07	**0**	0.02	0	0.1	**0**
Smoking	Never	NA	NA	NA	NA	NA	NA	NA	NA
	Previous	1.19	0.69	2.01	0.5216	1.52	0.78	2.92	0.2079
	Current	4.86	1.96	13.16	**0.001**	7.26	2.38	24.6	**0.0008**
Alcohol Use	None	NA	NA	NA	NA	-	-	-	-
	≤2/weeks	1.44	0.86	2.38	0.1575	-	-	-	-
	>2/weeks	0.82	0.44	1.49	0.524	-	-	-	-
Menopausal Status	Pre-menopause	NA	NA	NA	NA	NA	NA	NA	NA
	Post-menopause	0.32	0.17	0.57	**0.0001**	1.25	0.52	3	0.6155
T-Stage	T1	NA	NA	NA	NA	-	-	-	-
	T2	0.9	0.56	1.43	0.6579	-	-	-	-
	T3	1.29	0.59	2.75	0.5097	-	-	-	-
	T4	2.16	0.5	9.33	0.2863	-	-	-	-
	Ti	0.76	0.27	1.91	0.5805	-	-	-	-
Nodal Status	No	NA	NA	NA	NA	-	-	-	-
	Yes	0.91	0.6	1.38	0.6662	-	-	-	-
Receptor Status	HR+/HER2-	NA	NA	NA	NA	NA	NA	NA	NA
	TP	2.3	1.2	4.38	**0.0112**	2.72	1.16	6.43	**0.0214**
	TN	3.04	1.58	5.89	**0.0009**	3.22	0.51	22.56	0.2175
	HR-/HER2+	2.03	0.87	4.56	0.0905	2.33	0.33	18.55	0.4038
Previous Breast Cancer	No	NA	NA	NA	NA	NA	NA	NA	NA
	Yes	0.41	0.14	1.02	0.0782	0.35	0.08	1.21	0.125
Current Cancer Recurrent	No	NA	NA	NA	NA	-	-	-	-
	Yes	0.24	0.01	1.27	0.1729	-	-	-	-
Time from Diagnosis	0–1 years	NA	NA	NA	NA	NA	NA	NA	NA
	1–2 years	1.75	0.31	13.55	0.5413	5.3	0.61	117.47	0.1748
	2–3 years	1.6	0.28	12.39	0.6081	4.66	0.53	103.8	0.2124
	3–5 years	0.97	0.18	7.16	0.9691	3.02	0.37	64.08	0.3541
	5–10 years	0.64	0.12	4.75	0.6167	1.57	0.19	33.21	0.7054
	10+	0.75	0.11	6.43	0.7713	1.82	0.17	44.01	0.6478
Lumpectomy	No	NA	NA	NA	NA	NA	NA	NA	NA
	Yes	0.69	0.45	1.06	0.0891	3.09	0.84	12.45	0.0945
Mastectomy	No	NA	NA	NA	NA	NA	NA	NA	NA
	Yes	1.56	1.03	2.37	**0.0354**	3.35	0.89	13.64	0.0777
SN Biopsy	No	NA	NA	NA	NA	-	-	-	-
	Yes	1.89	1.08	3.47	**0.0309**	-	-	-	-
Axillary Dissection	No	NA	NA	NA	NA	-	-	-	-
	Yes	0.66	0.34	1.22	0.1975	-	-	-	-
Reconstruction	No	NA	NA	NA	NA	NA	NA	NA	NA
	Yes	2.31	1.36	3.93	**0.002**	1.25	0.56	2.76	0.5792
Endocrine Therapy Type	Neither	NA	NA	NA	NA	-	-	-	-
	Tamoxifen	0.95	0.5	1.81	0.8802	1.32	0.22	8.61	0.7625
	AI	0.39	0.22	0.7	**0.0016**	1.44	0.25	9.37	0.6903
	Both	0.76	0.4	1.44	0.3963	1.53	0.25	10.31	0.6498
Chemotherapy	No	NA	NA	NA	NA	NA	NA	NA	NA
	Yes	1.67	1.11	2.52	**0.015**	0.86	0.45	1.64	0.6419
Zoledronic Acid	No	NA	NA	NA	NA	NA	NA	NA	NA
	Yes	0.68	0.44	1.04	0.0762	0.76	0.42	1.37	0.3652
Radiation	No	NA	NA	NA	NA	-	-	-	-
	Yes	0.94	0.52	1.72	0.8238	-	-	-	-
Currently on Endocrine Therapy	No	NA	NA	NA	NA	-	-	-	-
	Yes	0.98	0.65	1.49	0.9225	-	-	-	-

NA refers to data “not available”.

**Table 9 cancers-17-00328-t009:** Odds ratios, confidence intervals, and *p*-values from bivariate and multivariate logistic regression models quantifying the association between demographic/clinical features and the likelihood of requiring close follow-up.

Variable	Bivariate Model					Multivariate Model			
	Odds Ratio		95% Confidence Interval		*p*-Value	Odds Ratio	95% Confidence Interval		*p*-Value
			Lower	Upper			Lower	Upper	
Age	0–45 years	NA	NA	NA	NA	NA	NA	NA	NA
	45–55 years	0.19	0.04	0.64	**0.0136**	0.16	0.03	0.57	**0.009**
	55–65 years	0.19	0.04	0.64	**0.0139**	0.18	0.04	0.68	**0.0181**
	65–75 years	0.11	0.02	0.36	**0.001**	0.1	0.02	0.37	**0.0014**
	75+	0.12	0.03	0.42	**0.0020**	0.13	0.03	0.48	**0.0047**
Smoking	Never	NA	NA	NA	NA	NA	NA	NA	NA
	Previous	1.43	0.86	2.36	0.1686	1.49	0.84	2.62	0.1662
	Current	2.35	0.97	5.9	0.061	2.18	0.84	5.79	0.1106
Alcohol Use	None	NA	NA	NA	NA	-	-	-	-
	≤2/weeks	1.3	0.79	2.12	0.2954	-	-	-	-
	>2/weeks	0.73	0.4	1.29	0.2898	-	-	-	-
Menopausal Status	Pre-menopause	NA	NA	NA	NA	NA	NA	NA	NA
	Post-menopause	0.47	0.26	0.85	**0.0125**	0.8	0.36	1.76	0.5737
T-Stage	T1	NA	NA	NA	NA	-	-	-	-
	T2	1.34	0.86	2.07	0.1901	-	-	-	-
	T3	1.8	0.85	3.81	0.1194	-	-	-	-
	T4	1.08	0.22	4.51	0.9161	-	-	-	-
	Ti	1.16	0.47	2.75	0.7424	-	-	-	-
Nodal Status	No	NA	NA	NA	NA	-	-	-	-
	Yes	0.97	0.65	1.44	0.8643	-	-	-	-
Receptor Status	HR+/HER2-	NA	NA	NA	NA	-	-	-	-
	TP	1	0.52	1.9	0.9897	-	-	-	-
	TN	0.98	0.5	1.89	0.9561	-	-	-	-
	HR-/HER2+	1.6	0.71	3.59	0.2537	-	-	-	-
Previous Breast Cancer	No	NA	NA	NA	NA	-	-	-	-
	Yes	0.77	0.34	1.65	0.5084	-	-	-	-
Current Cancer Recurrent	No	NA	NA	NA	NA	-	-	-	-
	Yes	0.66	0.14	2.42	0.5563	-	-	-	-
Time from Diagnosis	0–1 years	NA	NA	NA	NA	NA	NA	NA	NA
	1–2 years	1.05	0.18	6.18	0.9592	0.58	0.07	4	0.5767
	2–3 years	1.81	0.3	10.82	0.4962	1.16	0.14	8.14	0.8823
	3–5 years	0.59	0.11	3.3	0.5285	0.36	0.04	2.32	0.2791
	5–10 years	0.46	0.08	2.57	0.3559	0.23	0.03	1.46	0.1179
	10+	0.47	0.07	3.08	0.4155	0.26	0.03	2.02	0.2012
Lumpectomy	No	NA	NA	NA	NA	-	-	-	-
	Yes	0.73	0.49	1.1	0.135	-	-	-	-
Mastectomy	No	NA	NA	NA	NA	-	-	-	-
	Yes	1.42	0.95	2.12	0.0874	1.3	0.77	2.19	0.3298
SN Biopsy	No	NA	NA	NA	NA	-	-	-	-
	Yes	1.01	0.62	1.68	0.9547	-	-	-	-
Axillary Dissection	No	NA	NA	NA	NA	-	-	-	-
	Yes	1.03	0.58	1.8	0.9165	-	-	-	-
Reconstruction	No	NA	NA	NA	NA	NA	NA	NA	NA
	Yes	1.65	0.98	2.8	0.0599	1.15	0.56	2.35	0.7083
Endocrine Therapy Type	Neither	NA	NA	NA	NA	-	-	-	-
	Tamoxifen	1.25	0.66	2.39	0.4969	-	-	-	-
	AI	0.85	0.48	1.5	0.5642	-	-	-	-
	Both	1.52	0.81	2.89	0.1904	-	-	-	-
Chemotherapy	No	NA	NA	NA	NA	-	-	-	-
	Yes	1.38	0.93	2.03	0.1072	-	-	-	-
Zoledronic Acid	No	NA	NA	NA	NA	-	-	-	-
	Yes	0.8	0.54	1.2	0.2874	-	-	-	-
Radiation	No	NA	NA	NA	NA	-	-	-	-
	Yes	0.79	0.45	1.4	0.4232	-	-	-	-
Currently on Endocrine Therapy	No	NA	NA	NA	NA	-	-	-	-
	Yes	1.08	0.73	1.61	0.6922	-	-	-	-

NA refers to data “not available”.

## Data Availability

Data is contained within the article.

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
