# Peer review of "Risk Factors for Treatment Toxicity and High Side Effect Burden Among Breast Cancer Survivors: A Retrospective Chart Review"

_cancers, 2025, doi:10.3390/cancers17020328_

Round 1

Reviewer 1 Report

Comments and Suggestions for Authors

This is an extremely well written manuscipt. It is well done and 'flows' extremely well. A pleasure to review.

I only have some minor suggestions:

1.  why only capture data for three months why not more e.g. six months or a year and have more 'powerful' data?

2. on line 301 "clinical implications" first line"...in accordance with the latest guidelines.." spelling them out would be helpful or at least a reference.

However, great work.

Author Response

Comment 1: This is an extremely well written manuscipt. It is well done and 'flows' extremely well. A pleasure to review. I only have some minor suggestions. why only capture data for three months why not more e.g. six months or a year and have more 'powerful' data?

Response 1: Thank you for pointing this out. After reviewing the methods section again, we realized there was a typo in the data collection dates. We collected data from July 6, 2022 to September 30, 2023 (a year and 2 months). This has also been updated in the manuscript on line 34 of the abstract and line 294  of the Methods and Materials section.

Comment 2: on line 301 "clinical implications" first line"...in accordance with the latest guidelines.." spelling them out would be helpful or at least a reference.

Response 2: Thank you for this comment. We have added the specific guidelines, specifically the American Society of Clinical Oncology (ASCO) Breast Cancer Survivorship Guideline (2016), the National Comprehensive Cancer Network (NCCN) Breast Cancer Guideline (2024), as well as the National Cancer Institute (NCI) “Quality of Cancer Survivorship Care Framework”, by Nekhlyudov et al. (2019). The changes can be found on lines 547-551.

Reviewer 2 Report

Comments and Suggestions for Authors

The document addresses a significant yet often neglected dimension of cancer survivorship, exploring the elements that contribute to treatment-related toxicity and adverse effects in individuals who have survived breast cancer. The investigation is structured proficiently, employs a robust methodology, and presents actionable insights that could substantially enhance patient support post-recovery. However, there are possibilities for refinement to strengthen the scientific rigor and communication of the findings.

- The introduction effectively highlights the necessity of survivorship support. Nonetheless, it would be beneficial to add a brief note on the scarcity of research in the available literature regarding young and elderly women who have triumphed over breast cancer.

- The study framework is appropriate, yet it should clarify the reasoning behind choosing "two or more side effects" as the standard for defining a "high burden" and how this relates to clinical implications. 

- In discussion, please provide in-depth recommendations for tailoring survivorship support based on the findings. Advocate for prospective research initiatives to validate these findings, particularly concerning smoking and younger demographic groups.

Author Response

Comment 1: The document addresses a significant yet often neglected dimension of cancer survivorship, exploring the elements that contribute to treatment-related toxicity and adverse effects in individuals who have survived breast cancer. The investigation is structured proficiently, employs a robust methodology, and presents actionable insights that could substantially enhance patient support post-recovery. However, there are possibilities for refinement to strengthen the scientific rigor and communication of the findings.

- The introduction effectively highlights the necessity of survivorship support. Nonetheless, it would be beneficial to add a brief note on the scarcity of research in the available literature regarding young and elderly women who have triumphed over breast cancer.

Response 1: Thank you for this comment. We have added a brief note on the scarcity of research in the available literature regarding young and elderly women overcoming breast cancer (see lines 92-94).

Comment 2: The study framework is appropriate, yet it should clarify the reasoning behind choosing "two or more side effects" as the standard for defining a "high burden" and how this relates to clinical implications.

Response 2: Thank you for pointing this out. We have added the reasoning behind choosing “two or more side effects”. We chose two or more side effects through team consensus and from previous literature (lines 402-403). Furthermore, in the discussion, we have clarified that high side effect burden was defined as two or more side effects since 72.6% of the population had at least one side effect, and hence, two side effects were chosen to represent high side effect burden to better understand the potential risk to breast cancer survivors (see lines 577-580).

Comment 3: In discussion, please provide in-depth recommendations for tailoring survivorship support based on the findings. Advocate for prospective research initiatives to validate these findings, particularly concerning smoking and younger demographic groups.

Response 3: Thank you for this comment. We have added in-depth recommendations for tailoring survivorship support based on the findings, including specialized personal survivorship support and dedicated multidisciplinary teams (see lines 775-781 and lines 785-786). We have also added the need for prospective research initiatives to validate these findings (see lines 809-812).

Reviewer 3 Report

Comments and Suggestions for Authors

The manuscript presented for review is an interesting analysis of the Risk Factors for Treatment Toxicity and High Side Effect Burden among Breast Cancer Survivors in a follow-up center. The issue is of great importance especially in this era when amids increased survivability of Oncologic patients we are faced with an everincreasing need to address quality of life issues in both patients and in survivors.

The overall quality of tge manuscript is good and the structure is adequate.  My suggestions and concerns are as follows: 

1. There is absolutely no need for the very extensive presentation of the workings of the hospital. Please remove it since is out of context and of no interest.

2. There is a pie chart in text that looks very liw quality - the information can be presented in text or as a table 

3. I do not agree with the definition of survivorsip as starting from the diagnosis as stated in text. What is then the diference between patients and survivors?  Do the results include patients currently undergoing cancer treatment? Please clarify this aspect.

Author Response

Comment 1: The manuscript presented for review is an interesting analysis of the Risk Factors for Treatment Toxicity and High Side Effect Burden among Breast Cancer Survivors in a follow-up center. The issue is of great importance especially in this era when amids increased survivability of Oncologic patients we are faced with an everincreasing need to address quality of life issues in both patients and in survivors. The overall quality of tge manuscript is good and the structure is adequate.  My suggestions and concerns are as follows: There is absolutely no need for the very extensive presentation of the workings of the hospital. Please remove it since is out of context and of no interest.

Response 1: The authors would like to thank the reviewer for their kind comments. We have removed the information about the workings of the hospital (lines 113-125) but have kept the information pertaining to the patient population the clinic serves. Since we have a unique patient population that includes patients on endocrine therapy who come to the clinic a few months after the completion of active treatment, we believe it is important to clarify this population, as it may affect the results.

Comment 2: There is a pie chart in text that looks very liw quality - the information can be presented in text or as a table.

Response 2: Thank you for pointing this out. We have removed the pie chart (Figure 3). The information is presented in the text in lines 555-558. 

Comment 3: I do not agree with the definition of survivorsip as starting from the diagnosis as stated in text. What is then the diference between patients and survivors?  Do the results include patients currently undergoing cancer treatment? Please clarify this aspect.

Response 3: Thank you for this comment. The definition we have included in the manuscript is from the National Cancer Institute, and is the most recent definition of cancer survivorship. However, we did specify in the methods section that the study only included breast cancer survivors who have completed active treatment. Specifically, the inclusion and exclusion criteria in lines 294-297 states this:

Patients were included if they were older than 18 years, had completed treatment for at least one breast cancer at the time of extraction, and had attended at least one appointment at the Sunnybrook Breast Cancer Survivorship Clinic. Patients were excluded if they did not attend any appointments, had not had a confirmed diagnosis of breast cancer, or were undergoing surgery, chemotherapy, or radiation therapy for their first breast cancer at the time of data extraction. 

Round 2

Reviewer 3 Report

Comments and Suggestions for Authors

The authors have addressed my concerns adequately and the manuscript can be published in its current form.